# From nanohole to ultralong straight nano-channel fabrication in graphene oxide with swift heavy ions

Andrzej Olejniczak [1,2] & Ruslan A. Rymzhanov [2,3]

Porous architectures based on graphene oxide with precisely tailored nm-sized pores are attractive for biofluidic applications such as molecular sieving, DNA sequencing, and recognition-based sensing. However, the existing pore fabrication methods are complex, suffer from insufficient control over the pore density and uniformity, or are not scalable to large areas. Notably, creating vertical pores in multilayer films appears to be particularly difficult. Here, we show that uniform 6–7 nm-sized holes and straight, vertical nano-channels can be formed by simply irradiating graphene oxide (GO) films with high-energy heavy ions. Long penetration depths of energetic ions in combination with localized energy deposition and effective self-etching processes enable the creation of through pores even in 10 μm-thick GO films. This fully scalable fabrication provides a promising possibility for obtaining innovative GO track membranes.

The advantages of graphene oxide (GO), such as chemical tunability, flexible processability, and scalable low-cost synthesis, make it a desirable platform for designing porous architectures for nanofluidics[1–4] and energy storage[5–8]. In particular, single- and few-layer GO films with nm-sized holes and straight nanochannels are attractive for biomolecular separation[9], DNA sequencing[10,11], sensorics[12,13], and biorecognition molecule immobilization[14,15]. The essential demand for these applications is that the created pores be well-shaped, uniform in size, and obtainable in a reproducible manner[1,16]. The majority of developed methods are based on etching of the reactive (e.g., defective[7], highly oxidized[6,17,18], strained[19,20]) or catalyst-decorated[21–23] sites on GO sheets and, as such, typically offer only limited control over the porosity parameters and suffer from possible structure breakdown[12]. The etched GO sheets, once stacked into multilayer films, comprise a pore network consisting of randomly distributed in-plane holes and horizontal slit channels defined by the interlayer space[24,25]. Such a mixed pore system shows improved performance in, e.g., desalination and ion-selective transport[26,27], but due to undesirably long flux pathways and fouling with large molecules, it is not very effective for ultrafiltration, biomolecular sieving, and high-rate energy storage[24].

Among known techniques, only focused beams of charged particles[28–30] and guided-etching perforation[9] have demonstrated potential for fabricating vertical straight pores in few-layer GO and graphene that satisfy the uniformity and reproducibility criteria. Guided etching, applying self-organized block copolymer masks, can be scaled up to large areas but is a complex, multistep procedure and has not yet attained a pore size in the few nm range[9], i.e., the size at which confined nanofluidic phenomena emerge[31]. Patterning with focused beams of ions and electrons is free from these disadvantages; however, being an inherently serial approach, it is impractical for fabricating large multipore arrays. Based on these limitations, the need to develop inexpensive, effective, and relatively simple methods for the creation of well-defined nanoporous architectures in GO sheets remains of urgent importance.

Recently, Wang et al.[32] developed a breakthrough transformation of GeV-energy heavy-ion-irradiated polymers into ion-selective track membranes with ultranarrow highly dense pores that did not require chemical etching. However, pore formation still required some treatment (e.g., UV exposure, electromigration, extraction) to photodegrade and remove the decomposition products from the latent track.

[1]Faculty of Chemistry, Nicolaus Copernicus University, ul. Gagarina 7, 87-100 Torun, Poland. [2]Joint Institute for Nuclear Research, Joliot-Curie 6, 141980 Dubna, Moscow Region, Russia. [3]Institute of Nuclear Physics, Ibragimov St. 1, 050032 Almaty, Kazakhstan. ✉e-mail: aolejnic@chem.umk.pl

Here, we show that owing to efficient self-etching processes, swift heavy ions are capable of creating straight nm-sized pores in GO films of any usable thickness (from single layer to $\geq 10\,\mu m$) without the need for any postirradiation treatment. The pores not only are uniform and well-shaped but also, due to irradiation-induced defunctionalization, have a partially ordered periphery, providing high electrical conductivity. Through the use of GO obtained by two different methods and state-of-the-art Monte Carlo (MC)-reactive molecular dynamics (RxMD) modeling, we establish the relationships between the GO composition and the swift heavy ion (SHI) irradiation-induced defunctionalization and pore creation efficiency. Based on this, we show that nanoperforation of multilayer GO films fulfilling certain compositional requirements is also achievable with mid-electronic-stopping-power ions available in industrial-based cyclotron facilities.

## Results

### Fabrication of nanopores by SHI irradiation

Transmission electron microscopy (TEM) observation of the GO specimens irradiated with 710 MeV Bi ions provided direct experimental confirmation of nanohole creation in suspended single- and few-layer GO (Fig. 1a, b). In this case (710 MeV Bi ions), the initial energy deposition into the electronic subsystem was the highest. For few-layer (3-4) GO films, the sizes of the formed nanoholes were normally distributed, with a mean value of 5.95 nm and a standard deviation of 0.7 nm. The pores formed in single-layer GO were slightly larger (average diameter of ~7.5 nm) and less uniform in size (standard deviation of 1.35 nm). In addition, they were not as perfectly round shaped as the pores created in few-layer structures. The main reason for this behavior for single-layer GO is the lack of a supporting framework combined with the inhomogeneous local distribution of the oxygen functional groups attached to the graphene sheet—as known, in partially oxidized graphene, nanoscale regions with low and high oxygen group densities exist[33,34]. In the computational part of this paper, we will show that the number of carbon atoms removed from the lattice by an ion impact varies in proportion to the functionalization density as well that the internal layers of few-layer GO are stabilized by the outer layers, thus diminishing the strain, which in turn reduces cracking of the nanohole edge. An additional contribution to the pore size nonuniformity in single-layer GO arises from the difference in energy deposition (up to 2 times for graphene[35]) between the impact points of different electron densities; in multilayer GO, random stacking of sheets and interlayer energy exchange diminishes this detrimental effect.

Experimental evidence of the formation of ultralong and straight nanochannels was delivered by scanning electron microscopy (SEM) investigations of supported GO-Z (for sample abbreviations, see the section titled "GO synthesis") films irradiated with 710 MeV Bi ions (Fig. 2a–c). The surface SEM images revealed the presence of surface features with a well-defined crater form, i.e., hillocks with holes at their centers (Fig. 2c). The average sizes of the nanopores, measured at their bases and at the background (BG) level, are 10.4 and 5.4 nm, respectively. The hillocks are 27 nm in diameter on average. Cross-sectional SEM images show that the formed nanochannels are ca. 6.6–7 nm in diameter (Fig. 2b), and they extend over the whole GO film, ~ 11.5 $\mu m$ in thickness, including to its bottom surface (Fig. 2a).

Below, we consider the possibility of creating nanoholes and nanochannels with mid- to low-electronic-stopping-power ions. In the case of 167 MeV Xe ions, the electronic energy loss is ca. 0.64 that of 710 Me Bi ions (Supplementary Table 1). This electronic stopping power ($Se$) value is not sufficient to give rise to nanohole formation in suspended few-layer GO films. High magnification TEM images show that each ion impact creates a clearly visible spot, ca. 7–10 nm in size, of structurally modified material (Fig. 3a). We propose that these spots represent partially defunctionalized regions with a significant number of vacancies and extended structural defects. Since the reactivity of defective graphene is higher[36,37], these regions should be susceptible to chemical modification that will allow selective etching of the pores under mild conditions.

In contrast, the energy deposition of 167 MeV Xe ions is still sufficient for the formation of nanochannels in supported GO films, as confirmed by cross-sectional SEM images of a 1.5 $\mu m$-thick GO-Z film irradiated to a fluence ($\Phi$) of $6 \times 10^{11}$ ions cm$^{-2}$ (Fig. 3b, c). The SEM image of the film surface (Fig. 3d) shows the presence of crater-like features, similar to those observed in the case of the specimens irradiated with 710 MeV Bi ions. The hillocks have an average diameter of 25.4 nm; the average diameters of the nanopores at the base and background level are 10.2 and 5.1 nm, respectively. Note that the pores and protrusions formed on folded parts of GO flakes are larger (Fig. 3d). This is because for irradiation under grazing incidence, the energy deposition to GO lattice is higher. Analysis of the cross-sectional SEM images (Fig. 3c, c') shows that the channels are ca. 4–5.2 nm in diameter; i.e., their widths are reduced compared to those formed in 710 MeV Bi ion-irradiated films.

The low end of the examined $Se$ values is represented by the results obtained for 61 MeV V ions. Here, we failed to find surface features, such as nanoholes or protrusions, resulting from the impacts of individual ions (i.e., in the track nonoverlapping regime, Fig. 3e). However, a significant change in the surface morphology occurs with increasing ion dose (Fig. 3f). Initially, at low fluences, the surface becomes rougher, but the sheet structure of the GO flakes is preserved. At high doses ($\Phi = 6 \times 10^{12}$ ions cm$^{-2}$), we observe the formation of round-shaped protrusions with diameters ranging from 40 to 70 nm. They are clearly visible in tilt-view SEM (Fig. 3f) images. Their appearance might be due to the accumulation of gaseous products released during decomposition accompanied by sheet distortion resulting from partial structural ordering.

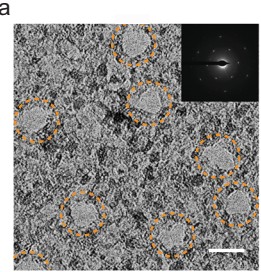
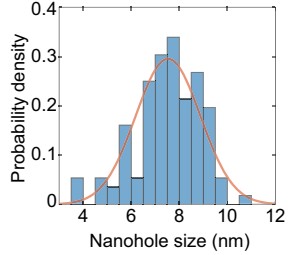
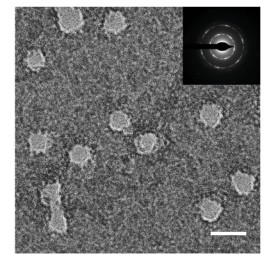
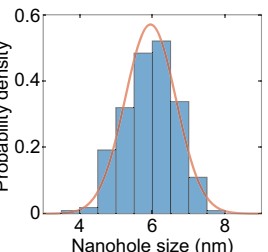

**Fig. 1 | Creation of nanoholes in suspended GO. a, b** TEM images and normalized pore size distributions of suspended GO specimens irradiated with 710 MeV Bi ions: **a** single-layer GO, ion fluence of $2 \times 10^{11}$ ions cm$^{-2}$, **b** 3–4-layer GO, ion fluence of $4 \times 10^{11}$ ions cm$^{-2}$. Insets show the corresponding selected area electron diffraction (SED) patterns. Orange lines represent normal distribution fits. The dashed circles mark the positions of the nanopores. Scale bars are 10 nm. Source data are provided as a Source Data file.

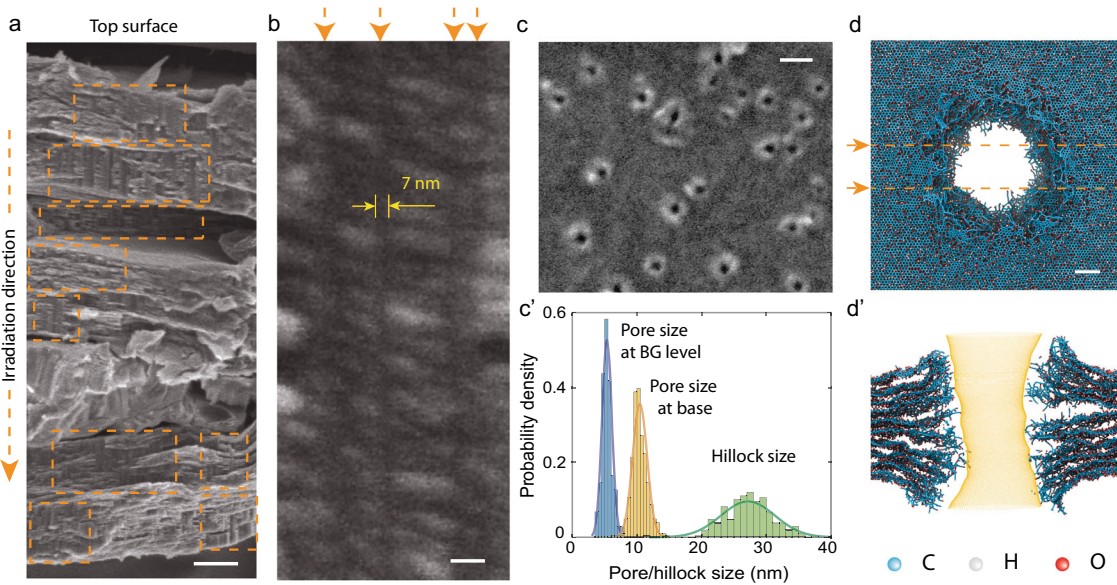

**Fig. 2 | Ultralong straight nanochannels formed in GO films. a, b** Cross-sectional SEM images of a GO-Z specimen (sputter coated with Pt/Pd) irradiated with 710 MeV Bi ions to a fluence of $1 \times 10^{11}$ ions $cm^{-2}$ at **a** low and **b** high magnification. **c** SEM image of the surface of the 710 MeV Bi ion-irradiated GO-Z film. **c′** Normalized distributions of pore and hillock sizes, determined as described in Supplementary Note 2. Source data are provided as a Source Data file. **d, d′** MC-RxMD simulations of nanochannel formation in multilayer GO: **d** top and **d′** cross-sectional view (shown on the same scale) along with the pore profile calculated with HOLE[52] using AMBER van der Waals radii. The scale bars are **a** 1 μm, **b** and **c** 20 nm, **d** 2 nm.

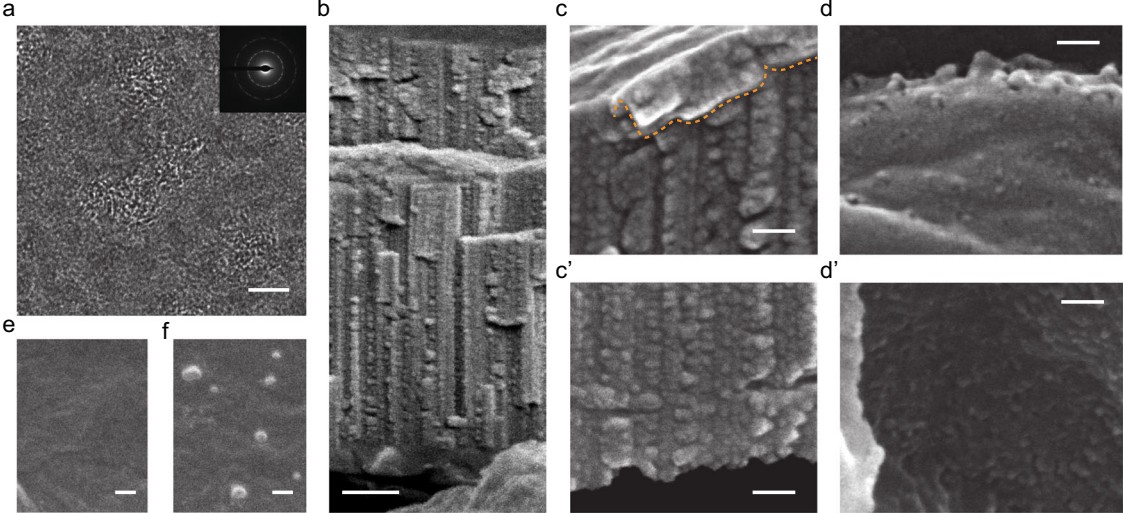

**Fig. 3 | Patterning of GO with mid- to low-electronic-stopping-power ions. a** TEM image of a suspended few-layer GO specimen irradiated with 167 MeV Xe ions to a fluence of $2 \times 10^{11}$ ions $cm^{-2}$. **b–d′** SEM images of a GO-Z specimen deposited on a nylon-66 membrane and irradiated with 167 MeV Xe ions to a fluence of $6 \times 10^{11}$ ions $cm^{-2}$ (sputter coated with Pt/Pd) **b, c** Cross-sectional images at **b** low and **c, c′** high magnification. The bent surface layer of the GO film is marked with orange dashed lines. **d, d′** SEM images of the **d** top and **d′** bottom surface of the GO film. **e, f** SEM images of GO-Z films irradiated with different doses of 61 MeV V ions. The scale bars are **a** 5 nm, **b** 200 nm, **c–f** 50 nm.

## Influence of the GO chemical composition on SHI irradiation damage and defunctionalization

To partially explain the influence of the structure and composition of GO on transformations induced by SHI bombardment, we compare GO-Z and GO-M films obtained by two different methods (Fig. 4a), i.e., conventional synthesis with Hummers reagents (GO-Z) and synthesis without $NaNO_3$ and under conditions facilitating exfoliation (GO-M). The initial samples were chosen so that their C/O atomic ratios, determined via X-ray photoelectron spectroscopy (XPS), were closely similar ($2.52 \pm 0.07$). A closer inspection of the Fourier transform infrared (FT-IR) spectra of the initial films (Fig. 4b and Supplementary Fig. 8), however, revealed significant differences in their chemical compositions. Compared to GO-M, the following differences in the FT-

IR spectra of GO-Z are most noticeable: (i) the band due to epoxy groups (1254 $cm^{-1}$) is less intense, (ii) the band at 1586 $cm^{-1}$ is broader due to the higher contribution of carbonyl groups, and (iii) the band at 1410 $cm^{-1}$, due to deformation C−OH vibration of carboxyl groups, is more pronounced. Such a picture is also confirmed by the C 1s core level spectra (Supplementary Fig. 9) and suggests that the GO-Z sample has more edge groups (mainly carbonyl and carboxyl), while GO-M has more basal-plane groups (mainly epoxy). The different distributions of oxygen-containing groups are related to the smaller flake size and possibly more defective carbon lattice of GO-Z.

The evolutions of the FT-IR and C 1s core level spectra of GO-M and GO-Z irradiated with different doses of 167 MeV Xe ions are shown in Fig. 4b and Supplementary Fig. 9, respectively. With increasing ion

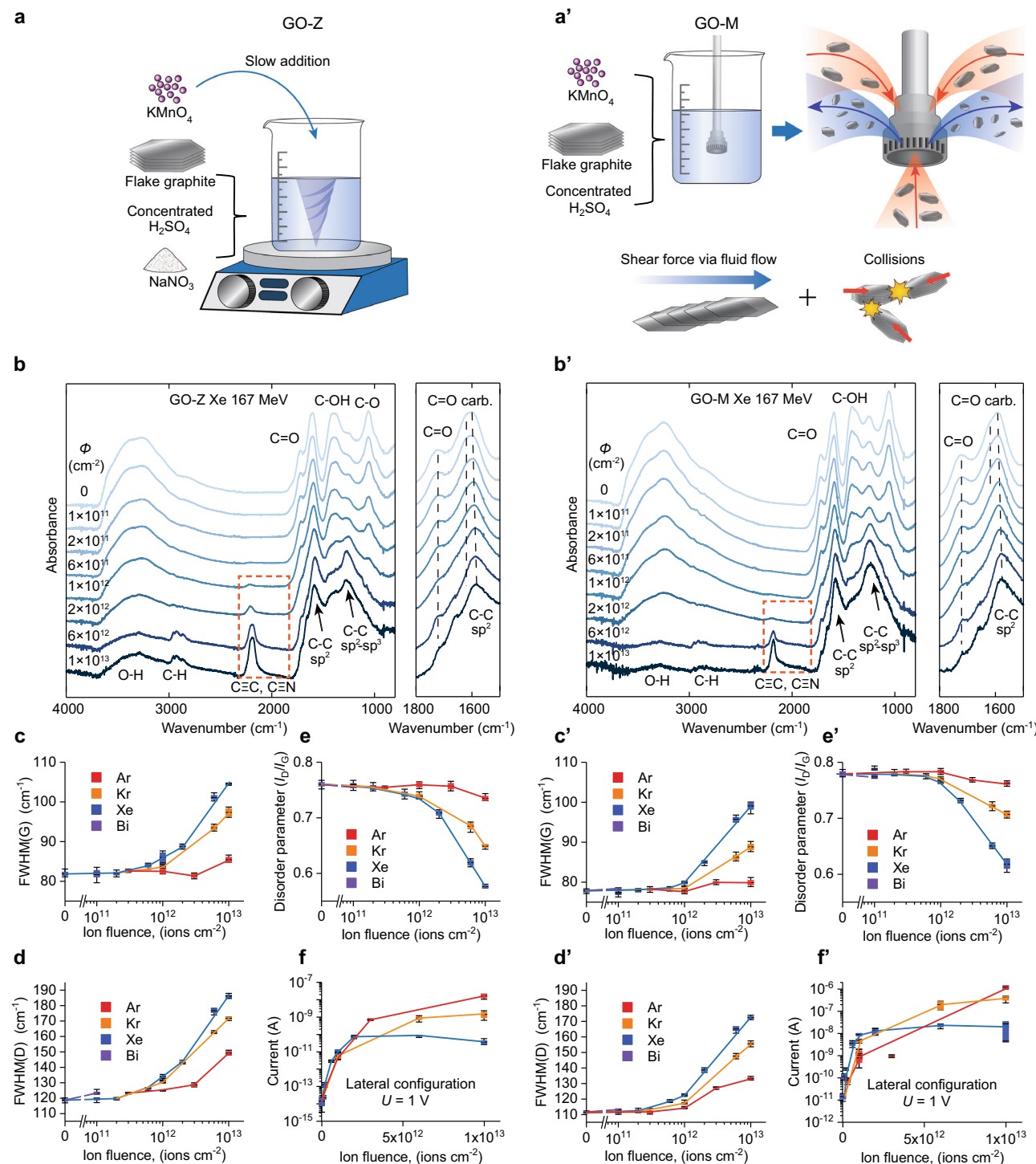

**Fig. 4 | Influence of the chemical composition of GO on SHI-irradiation-induced damage and defunctionalization. Left column** GO-Z films obtained by a conventional procedure with Hummers reagents; **right column** GO-M films obtained under conditions facilitating graphene exfoliation and without the use of NaNO₃. **a**, **a'** Schematics of GO-Z and GO-M preparation. **b**, **b'** FT-IR spectra of specimens irradiated with different fluences of 167 MeV Xe ions. The orange boxes mark the spectral region of conjugated C≡C bond stretching. **c**, **c'**–**e**, **e'** Fluence-dependent evolution of the Raman spectral parameters. **c**, **c'** FWHM of the G-band, **d**, **d'** FWHM of the D-band, **e**, **e'** D- to G-band intensity ratio (disorder parameter) for GO films irradiated with ions of different electronic stopping powers. **f**, **f'** Variations in the in-plane conductivities (as the current recorded at a $U$ of 1 V) for GO films irradiated with ions of different electronic stopping powers. In the plots: center line, mean value; box boundary, 95% confidence interval; whiskers, minimum and maximum. Source data are provided as a Source Data file.

fluence, the bands due to oxygen functionalities progressively disappear in both the FT-IR and XPS spectra. Eventually, at sufficiently high ion fluences, the FT-IR spectra show signals originating mainly from carbon vibrations. Namely, the peak at 1620 cm⁻¹ becomes narrower, and its maximum shifts toward lower wavenumbers characteristic of sp² C=C clusters. The change in the relative intensity of the signal at ~1250 cm⁻¹ is complex; it initially decreases up to a dose of $3 \times 10^{12}$ ions cm⁻² and then becomes more intense. Such behavior results from the overlap of the epoxy group vibration with a new band that appears at ~1250 cm⁻¹. This new band is attributed to sp²−sp³ C

clusters[38,39] and proves partial amorphization of the C lattice at high doses. The weak complex band at ~2850 cm$^{-1}$ corresponds to a C−H stretching mode; it indicates a possible building-in of H atoms originating from decomposition of hydroxyl groups.

The band with a maximum at ~2200 cm$^{-1}$ and a tail extending to 1900 cm$^{-1}$ is characteristic of conjugated C≡C bond stretching ($\alpha$- and $\beta$-modes, respectively)[40,41] and confirms the presence of $sp$-hybridized carbon ($sp$-C). Since the main ~2200 cm$^{-1}$ peak, especially for GO-M, is well-defined and rather narrow, a contribution from C≡N groups resulting from the incorporation of nitrogen into reactive sites is possible. This interpretation is supported by the evolution of the N 1$s$ core-level spectra and changes in the N/C atomic ratio of GO-Z films irradiated with different fluences of 167 MeV Xe ions (Supplementary Fig. 10).

In the section on MC-RxMD modeling, we will show that the formation efficiencies of $sp$-C chains and extended defects, including nanoholes, are mutually correlated. Thus, based on the intensity evolution of the ~2200 cm$^{-1}$ peak, we compare the damage efficiencies for GO-M and GO-Z. For GO-Z irradiated with 167 MeV Xe ions, the ~2200 cm$^{-1}$ band emerges at a lower dose ($6 \times 10^{11}$ vs. $2 \times 10^{12}$ ions cm$^{-2}$ for GO-M) and is much more intense for high-dose-irradiated samples. Similarly, for samples irradiated with lighter ions (e.g., 107 MeV Kr, Supplementary Fig. 11b), the ~2200 cm$^{-1}$ band is more intense for GO-Z than GO-M irradiated with the same fluence. For GO-M, the 2200 cm$^{-1}$ band is absent for 46 MeV Ar-ion-irradiated samples in the investigated range of fluences (up to $1 \times 10^{13}$ ions cm$^{-2}$), and for GO-Z it appears only at the highest fluence (Supplementary Fig. 11a). These results are in line with the SEM observations and confirm that (i) the formation of nanoholes does not occur at low electronic stopping powers and (ii) for ions with high $Se$ values, the formation of nanopores is less efficient for GO-M (Supplementary Fig. 12).

Additional insights into structural changes occurring upon SHI irradiation are provided by Raman spectroscopy. The Raman spectra of the initial GO-Z and GO-M are closely similar. A careful analysis of the deconvoluted spectra shows, however, that the spectral bands of GO-Z, especially the D-band, are somewhat broader (full width at half maximum (FWHM) of the D-band: 110 cm$^{-1}$ for GO-M vs. 120 cm$^{-1}$ for GO-Z, Fig. 4d). In addition, the disorder parameter, defined as the intensity ratio of the D- to G-band ($I_D/I_G$), is slightly lower for GO-Z (0.75 vs. 0.77 for GO-M, Fig. 4e). Since both materials lie at stage II along the amorphization trajectory of Robertson and Ferrari[42], these differences indicate that GO-Z is shifted toward a more disordered structure (i.e., has a smaller domain size). This can be due to more abundant structural defects within GO flakes and their smaller sizes. This interpretation is in agreement with the fact that GO-Z was obtained under more severe conditions, whereas the synthesis procedure for GO-M was oriented toward facilitating exfoliation of the graphene flakes.

The changes in Raman-derived spectral parameters for GO-Z and GO-M irradiated with different fluences of Xe, Kr, and Ar ions are shown in Fig. 4c−e. For all ions, irradiation to high fluences leads to broadening of the Raman D- and G-bands and a decrease in the $I_D/I_G$ ratio. These spectral changes prove that the GO structure is damaged by the ions and eventually becomes partially amorphized. Structural damage is more significant for ions with higher $Se$ values (e.g., 167 MeV Xe). For 46 MeV Ar ions with low electronic stopping power, the damage is significantly suppressed, and at low doses (up to 10$^{12}$ ions cm$^{-2}$), the Raman spectra exhibit signatures of structural ordering. The ordering results from structural recovery in the track halo predominating over radiation damage at the track core (see the stochastic core-halo model in Supplementary Fig. 13) and was studied in detail in ref. [43,44]. For all ions, the decrease in the $I_D/I_G$ ratio and broadening of the spectral features are more significant for GO-Z than for GO-M. This confirms that GO-Z is more susceptible to SHI-irradiation damage, in agreement with the FT-IR results.

Because of the formation of structurally ordered nanoscale regions with improved electrical transport properties, irradiation leads to significant enhancement of the overall conductivity of the films (Fig. 4f). In the investigated range of fluences, the following trends are visible. For Xe ions, the conductivity reaches a maximum at a fluence of ca. $2 \times 10^{12}$ ions cm$^{-2}$ and then gradually degrades; in the case of Kr ions, the conductivity increases, reaching saturation at a fluence of ca. $10^{13}$ ions cm$^{-2}$, whereas for Ar ions, the conductivity continuously improves. The rate of the initial improvement in the conductivity increases with increasing electronic stopping power, i.e., it is the most rapid for Xe ions and the slowest for Ar ions. The maximum enhancement in the conductivity follows a reverse tendency with $Se$, i.e., it is the highest for samples irradiated with Ar ions, reaching ~5 orders of magnitude; for samples irradiated with Kr ions, the enhancement is ca. 1–1.5 orders of magnitude lower, and in turn for Xe ions, it is yet another order of magnitude lower. These relations result from different changes in the recovery and damage efficiencies with electronic stopping power and can be semiquantitatively explained by a stochastic model assuming structural damage in the core and recovery in the halo (Supplementary Fig. 13). Since for low-$Se$ ions, the damage is suppressed to a greater extent than the recovery, a saturation phase occurs at higher fluences, and the maximum achievable conductivity is higher. The fact that conductivity improvement also occurs for high-$Se$ pore forming ions proves that the pore edges are locally defunctionalized and electrically conductive. The partial recovery of the graphitic structure at the frontier regions of the created nanoholes was confirmed by TEM and electron energy loss spectroscopy (EELS) (Supplementary Fig. 14). Such local structural restoration occurs due to the unique ability of swift heavy ions to modify the materials within the nanometer-sized cylinders around the ion trajectory (Supplementary Fig. 13).

A comparison of the in-plane conductivities of the pristine films shows that the conductivity of the GO-Z specimen is much worse than that of GO-M (Fig. 4f, f'). Similarly, for samples irradiated to the same fluences, the conductivities of GO-Z films are always lower than those of GO-M. This behavior as well as the higher damage vulnerability of GO-Z is not surprising in view of the structural characteristics of the initial films. As is known, a decreased flake size and the presence of defects are factors affecting the energy dissipation and transport properties. The presence of defects is also known to decrease the barrier for radiation damage. Another important factor is that GO-Z has abundant carbonyl moieties that have high thermal stability. Because of this, they can survive in the track halo periphery and reduce the defunctionalization efficiency. Removal of carbonyl species is expected to occur in the highly heated track core region. Since their decomposition proceeds via a mechanism involving reactions with adjacent C atoms[45,46], lattice etching in the track core is enhanced.

## MC-RxMD modeling of pore formation and structural recovery

To gain insight into the chemical and structural transformations of GO induced by energetic heavy ions, we simulated the impacts of a single 167 MeV Xe projectile on GO structures of different initial compositions. Since the simulations were carried out with periodic boundaries for in-plane directions, we limited our considerations to three sets of models, initially containing only epoxy and hydroxyl groups, with the compositional parameters varied, as shown in Table 1 and Supplementary Table 2. In the first set, the initial epoxy-to-hydroxyl group ratio was kept constant at 1, and the initial substitution $C_{sub}^{init}/C_{unsub}^{init}$ was varied among 3, 1, and 0.6. In the second set, the portion of C atoms substituted with hydroxyl groups was set to 0.25 (C/OH = 4), and the hydroxyl-to-epoxy group ratio was changed to 1, 0.5, 0.2, and 0, i.e., the last supercell in the series contained only OH groups. Here, the structure denoted $C_6H_{1.5}O_3$ is common to both series. Finally, the last series contained only epoxy groups, and the initial substitution

**Table 1 | Compositions of single-layer GO models studied with MC-RxMD**

| Formula | C:epoxy:OH | Initial | | Relaxed | | After ion passage | | |
|---|---|---|---|---|---|---|---|---|
| | | C:H:O | $C_{sub}/C_{all}$ (%) | C:H:O | $C_{sub}/C_{all}$ (%) | C:H:O | $C_{sub}/C_{all}$ (%) | $C_{rem}/C_{tot}$ (%) |
| Constant epoxy/OH ratio of 1.0, varied substitution ratio | | | | | | | | |
| $C_6H_{1.5}O_3$ | 4:1:1 | 2.00:0.50:1 | 75 | 2.05:0.48:1 | 69.5 | 2.17:0.42:1 | 66.6 | 4.00 |
| $C_6H_{1.0}O_2$ | 4:0.66:0.66 | 3.01:0.50:1 | 50 | 3.06:0.49:1 | 45.1 | 3.21:0.45:1 | 43.4 | 1.77 |
| $C_6H_{0.75}O_{1.5}$ | 4:0.5:0.5 | 4.02:0.50:1 | 37.5 | 4.04:0.49:1 | 35.0 | 4.26:0.46:1 | 33.5 | 0.94 |
| Constant C/OH ratio of 4.0, varied C/epoxy ratio | | | | | | | | |
| $C_6H_{1.5}O_{2.25}$ | 4:0.5:1 | 2.66:0.66:1 | 50 | 2.70:0.65:1 | 45.6 | 2.89:0.61:1 | 42.7 | 1.91 |
| $C_6H_{1.5}O_{1.8}$ | 4:0.2:1 | 3.33:0.83:1 | 35 | 3.35:0.83:1 | 32.9 | 3.64:0.79:1 | 30.2 | 0.84 |
| $C_6H_{1.5}O_{1.5}$ | 4:0:1 | 4.00:1.00:1 | 25 | 4.02:1.00:1 | 22.5 | 4.40:0.98:1 | 20.5 | 0.29 |
| Only epoxy groups (no OH) | | | | | | | | |
| $C_6O_{1.5}$ | 4:1:0 | 4.00:0.00:1 | 50 | 4.00:0.00:1 | 50 | 4.05:0.00:1 | 48.8 | 2.02 |
| $C_6O_{1.125}$ | 4:0.75:0 | 5.33:0.00:1 | 37.5 | 4.00:0.00:0.75 | 37.5 | 5.41:0.00:1 | 36.5 | 1.45 |
| $C_6O_{0.75}$ | 4:0.5:0 | 8.00:0.00:1 | 25 | 4.00:0.00:0.5 | 25 | 8.12:0.00:1 | 24.4 | 0.60 |

$C_{rem}/C_{tot}$ represents the fraction of C atoms removed from the GO layer due to single 167 MeV Xe ion impact;
$C_{sub}/C_{all}$ denotes the fraction of substituted C atoms.

$C_{sub}^{init}/C_{unsub}^{init}$ was varied among 1.0, 0.6, and 0.33. For the notation used throughout this section, please refer to Supplementary Note 3.

During initial cell equilibration, partial defunctionalization occurred for some cells; the heteroatomic species were released or weakly bonded to functional groups. As a result, especially for heavy OH-substituted structures, the attained substitution was somewhat lower than the assumed substitution (Table 1). Because of this, the extent of structural modification due to ion passage was evaluated by relating the compositional parameters of ion-irradiated cells to those of relaxed cells as a reference.

The structures obtained after the passage of a 167 MeV Xe ion are shown in Fig. 5a and Supplementary Fig. 15. For all GO models, irradiation resulted in the creation of a nanohole in the track core region. The average sizes of the nanoholes range from 2.5 to 7.6 nm, as determined from the atomic density drop in the radial distribution profiles (Fig. 5b and Supplementary Fig. 15b). A primary factor influencing the removal of C atoms from the lattice is the substitution degree ($C_{sub}/C_{all}$) (Fig. 6a). The efficiency of removing C atoms upon ion passage, expressed as $C_{rem}^{irrad}/C_{tot}$, increases approximately linearly with $C_{sub}/C_{all}$ in the relaxed cells. Some influence of the functionality type on the removal of C atoms ($C_{rem}^{irrad}/C_{all}$) is also visible but is not as pronounced as in RxMD simulations considering thermal treatment of a whole GO cell[47,48].

A major process in the track halo is defunctionalization. Generally, the absolute difference in the substitution degree between relaxed and irradiated cells, $\Delta(C_{sub}/C_{all})$, increases with $C_{sub}^{relax}/C_{all}$ (Fig. 6b). This trend is also visible locally in the radial distributions of defunctionalized C atoms (Supplementary Fig. 16) and is explainable by different densities of heteroatoms in the affected zone. Compared with C atom removal, the defunctionalization is influenced by the group type composition to a much greater extent. For structures containing only epoxy groups, the decrease in the functionalization density, expressed by $\Delta(C_{sub}/C_{all})$, is lower than that for those structures with an initial epoxy/hydroxyl ratio of 1; these structures in turn are less prone to defunctionalization than $C_6H_{1.5}O_{1.5}$ containing only hydroxyl groups.

Since the actual substitution varies in different structures, even more direct trends are obtained by plotting the relative dependencies (Figs. 5b and 6c). In Fig. 6c, $\Delta(C_{sub}/C_{all})$ is divided by the substitution degree for relaxed cells ($C_{sub}^{relax}/C_{all}^{relax}$) and plotted as a function of $C_{sub}^{relax}/C_{all}$. This obtained relative change in the functional group density, $\Delta(C_{sub}/C_{all})/(C_{sub}^{relax}/C_{all}^{relax})$, is almost constant for the sets of cells with epoxy/OH ratios of 1:0 and 1:1. For structures with varied epoxy/OH ratios, it linearly decreases with $C_{sub}^{relax}/C_{all}$, i.e., with increasing portion of epoxy groups.

Figure 5b presents the local normalized substitution degree ($C_{sub}/C_{all}^{\#}$), plotted along with the normalized atomic densities as radial distributions around the ion impact point. The region constrained by a drop in the C atom density and a decline in the $C_{sub}/C_{all}^{\#}$ ratio represents a zone of decreased functional group density. This zone is narrow (~2–3 nm in width) for structures containing only epoxy groups. With increasing portion of hydroxyl groups in the GO cell, low values of $C_{sub}/C_{all}^{\#}$ are attained over a broader region. In these cases, especially for $C_6H_{1.5}O_{1.8}$ and $C_6H_{1.5}O_{1.5}$, track halos with reduced functional group densities are clearly visible in Fig. 5a. For some structures, in the zone of decreased functionalization density, a sudden increase in $C_{sub}/C_{all}^{\#}$ is visible in the radial region representing the edge of the nanohole. This effect is related to back-oxidation, i.e., incorporation of oxygen into the carbon backbone due to reactions between irradiation-generated transient species and reactive C sites.

Several studies have explored the possibility of creating graphene quantum dots (Q-dots) by locally defunctionalizing graphene derivatives with SHIs[43,49]. In ref. [43], low-temperature transport in 46 MeV Ar-ion-irradiated GO films was described by Efros-Shklovskii variable range hopping, highlighting the confinement of wavefunctions within the recovered $sp^2$-C domains. With increasing ion fluence, the authors observed an increase in the localization length; however, the improvement was rather moderate. Similarly, for SHI-irradiated fluorinated graphene, the formation of only small, 1–3 nm-sized, Q-dots was observed in the track halo[49]. The present RxMD study brings insights into these experimental results, showing that for GO samples with a substantial portion of epoxy groups (practical case), the structural recovery is rather moderate. More importantly, the process is related to the creation of small graphene domains and/or expansion of the already existing domains. Such regions are visualized in Supplementary Fig. 17, showing the carbon backbones of the irradiated structures, with C atoms colored according to their hybridization type. Here, the recovered $sp^2$-C atoms are highlighted with lighter color. Highly effective ordering is found only for $C_6H_{1.5}O_{1.5}$, which is substituted with hydroxyl groups. In this case, the ordered region, with only a few defects and residual functionalities, is close to the hypothetical dot-antidot system, as proposed in ref. [43].

We should mention that defunctionalization does not automatically lead to restoration of a perfect graphene structure. A careful analysis shows that the recovered C atoms can have a bonding configuration (i.e., hybridization state) other than $sp^2$ (Supplementary Fig. 18). Among them, the $sp$-hybridized C atoms, forming carbyne chains, are the most abundant. The formation efficiency of $sp$-C chains

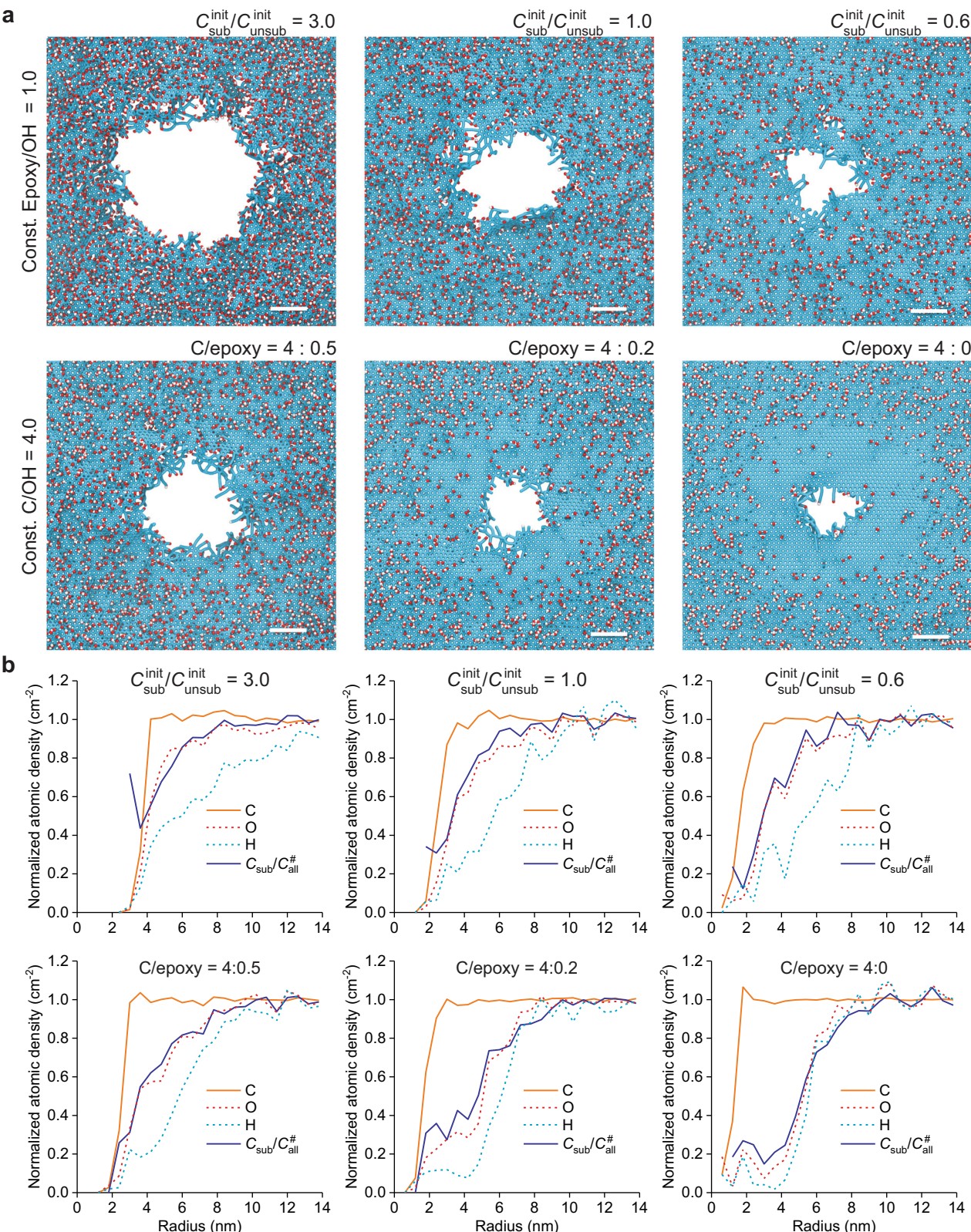

**Fig. 5 | MC-RxMD modeling of 167 MeV Xe ion passage through single-layer GO of different compositions. a** Final structures 0.1 ns after projectile passage; the scale bar is 2 nm. **b** Respective radial distributions of normalized atomic densities and normalized local substitution degrees ($C_{sub}/C_{all}^{\#}$).

is the highest in the vicinity of the nanohole (Supplementary Figs. 16 and 17) and is directly proportional to the number of removed C atoms and thus to the $C_{sub}^{relax}/C_{all}$ ratio. Based on atomistic representations, at least a fraction of the created $sp$-C chains is expected to be stable (e.g., those joining torn lattice edges[50]), thus explaining their experimental

detection here and in ref. [43,44]. Unterminated $sp$-C chains (i.e., those with dangling bonds) and oxygen end-capped chains are considered highly reactive. Upon exposure to air, the latter are supposed to incorporate both N and O, and the former (oxo forms) undergo sequential shortening[51].

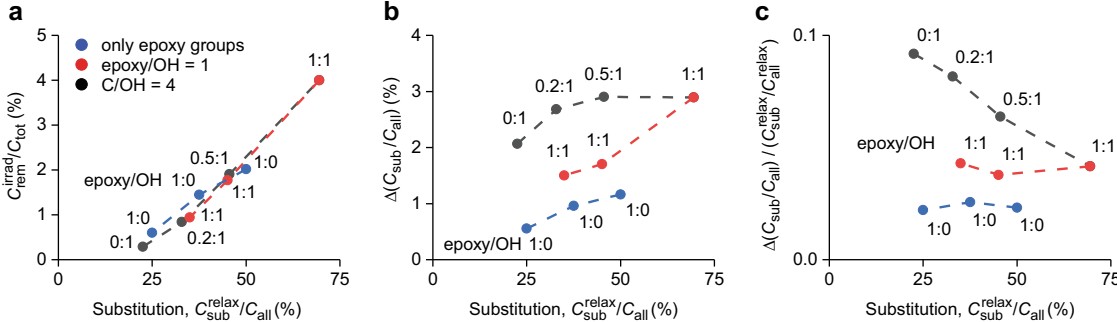

**Fig. 6 | Efficiencies of SHI-induced damage and defunctionalization of GO with different initial compositions. a** Removal of C atoms from the lattice and **b** absolute and **c** relative changes in the substitution degree as functions of the initial substitution degree ($C_{sub}^{relax}/C_{all}$). Source data are provided as a Source Data file.

Below, we discuss the influence of GO film thickness on pore size and shape symmetry. Here, we should distinguish different scenarios for single-, few-, and multilayer GO. For both single- and few-layer structures, the formed reactive intermediate species immediately escape to vacuum (Supplementary Movies 1 and 2). Upon ion impact, the unsupported lattice of single-layer GO undergoes significant distortion, which leads to the appearance of cracks at the nanohole edge and generally enlarges the nanohole size. For the 3-layer structure, upon ion impact, the outer layers temporarily expand outward, whereas the internal layer stays in place (Supplementary Movies 2 and Supplementary Fig. 19a). A similar stabilization effect occurred for other few-layer structures, resulting in a more perfect nanohole shape and narrowing the size of the nanohole created in the internal layers. The smaller nanohole size in few-layer than single-layer GO was confirmed by fitting the spherical vdW probe calculated with HOLE[52] (Supplementary Fig. 19b, c) and is in agreement with the experimental data (Fig. 1).

Simulations of 167 MeV Xe ion passage through multilayer GO were carried out for a supercell consisting of 10 single GO sheets; the results are presented in Fig. 1d. Based on the radial distribution profile, for the highest functionalization density ($C_6H_{1.5}O_3$), the width of the nanochannel in the central, narrow parts is ~ 9 nm, i.e., 15–18% larger than the width of the nanoholes created in single-layer GO with the same initial composition. This difference is caused by etching of the nanochannel edges by reactive species formed due to irradiation (Supplementary Movie 3). The analysis of evolved products allowed us to identify abundant quantities of CO, $H_2O$, $H_2O_2$, HO and its water adducts. These species are known to act as activation agents for etching pores in graphene and active carbons.

The cross-sectional view shows that nanochannel ends with a crater-like profile formed at the surface. This profile closely reassembles the surface features observed in the SEM images of 710 MeV Bi and 167 MeV Xe ion-irradiated Z-GO films (Fig. 2c). We attribute the crater-like shape to the evolution of low-molecular-weight species combined with lattice expansion due to the sp3-to-sp2 hybridization change of C atoms within the defunctionalized zone. As seen in the perspective projections of single-layer GO structures, the defunctionalized regions are bulkier and thus become curved with respect to the surface normal (Supplementary Fig. 20).

Finally, we compare the results of the MC-RxMD simulations with the experimental data. According to the present experiments, Xe ions of 167 MeV energy are capable of creating nanochannels in GO thin films but cannot drill nanoholes in unfolded suspended few-layer GO. In this case, only extended defects are created; formation of nanoholes is possible for ions with higher stopping power, e.g., 710 MeV Bi ions with a 1.56 times higher $S_e$ value. In contrast, the present MC-RxMD simulations show that each impact of a 167 MeV Xe ion creates a nanopore in single-layer GO, even for structures with low

functionalization density. This discrepancy is mainly caused by neglecting secondary electron emission (SEE) in our MC calculations and requires a brief discussion. The MC-complex dielectric function (CDF) approach treats the GO layer as a thin slice of bulk material, where all particles generated by ion passage lose their energy within this layer under periodic boundary conditions. While this approach is justified for layers sandwiched inside thick GO films, it overestimates the energy deposition in the near-surface region and sub-nm thin free-standing GO. This is because a fraction of electrons that overcome the work function and Coulomb barriers escape from the surface, carrying away a part of the deposited energy.

Accurate estimation of the SEE from 2D materials via MC-CDF requires accounting for the Coulomb interaction and is a topic of intensive study. As a first approach, we used data published for graphene, estimating that the fraction of energy emitted due to SEE constitutes 50% of the energy initially deposited by the Xe ion of the same velocity[35]. Using this SEE-corrected value, we compare the damage extent for GO structures having the same C/O ratio as our samples (~2.5) and a typical OH/epoxy composition (2:1)[33,53]. The obtained results (Supplementary Fig. 21) show that the reduction in energy deposition due to SEE is high enough to reduce the damage in the track core to the extent that only small pores, 0.4–0.9 nm in size, are created. The partially defunctionalized, sp2-C rich region around the ion trajectory is ca. 8–10 nm in diameter, in good agreement with the experimental case.

## Discussion

We demonstrate the fabrication of nanoholes and ultralong straight vertical nanochannels in GO by SHI irradiation. Since pore formation is assisted by self-etching processes, proceeding with the gasification of C atoms, it does not require any chemical or postirradiation treatment. The pores can be drilled in specimens of basically any usable thickness, from sub-nm single-layer free-standing sheets to ten-µm-thick GO films, thus allowing optimal permeability-selectivity performance in separation-oriented applications.

Starting from 3–4-layer-thick GO, the pores have a perfectly round geometry and a narrow size distribution in the few-nm range (~6 nm), highly desired for modern biofluidic technologies and electrochemical energy storage (for possible applications see Supplementary Note 1). The fabrication method provides a high density of pores (>10^11 vs., e.g., ~10^9 cm^−2 for classical track membranes[54]) and precise control over their density. Our MC-RxMD simulations and irradiation experiments predict that the pore size can be tuned by adjusting the initial functionalization density, group type composition, and electronic stopping power ($S_e$). The pore periphery is electrically conductive due to heaving partially restoring the graphitic structure and bears nitrile groups. The last feature opens the door to tailoring the channel sieving and sensing properties via rich functionalization chemistry[55].

The relationships between the composition and SHI-irradiation-induced transformations are important for achieving the desired behavior of GO films during nanopatterning. If the creation of nanopores is of interest, then the starting material should have a high oxidation degree, with abundant doubly bonded oxygen groups and some structural defects. A more perfect structure with a high portion of hydroxyl groups is preferred for obtaining optimum structural recovery and high electrical conductivity.

Based on well-established SHI irradiation technologies, the present fabrication technique can easily be scaled up, to large-area mass production (Supplementary Fig. 22), and down, to single-ion track devices.

## Methods

### GO synthesis

GO samples, denoted GO-Z and GO-M, were prepared from expandable graphite by two different modifications of the Hummers method. GO-Z was obtained according to the general procedure of Zhang et al.[56], which applies a similar set of reagents as the original Hummers method (Fig. 4a), followed by extensive purification by dialysis. The details of the synthesis and purification are given in[57]. The second type of sample, denoted GO-M, was obtained from Graphenea as described in[58]. Here, the synthesis was carried out without $NaNO_3$ and under high-shear mixing, facilitating exfoliation (Fig. 4a'). The obtained samples were characterized with respect to the C/O ratio, number of carbon layers and lateral dimensions, according to the recommendations in[59]. Remarks on the synthesis of GO and the obtained flake sizes are provided in Supplementary Note 4.

### Preparation of GO films

GO suspensions were diluted with water to a final concentration of 0.2 mg/ml, sonicated for 10 min with a pulsed ultrasonic processor, and centrifuged at $4000 \times g$ to remove residual particles and thick and/or weakly oxidized flakes. Then, the suspensions were vacuum-filtered on a membrane filter; the formed films were extensively washed with deionized water, dried, and peeled off from the membrane if required.

### Fabrication of nanopores in GO by SHI irradiation

To create nanopores, GO films were irradiated under vacuum with 710 MeV Bi ions to fluences of $10^{10}$ and $10^{11}$ ions cm$^{-2}$ at the U-400 cyclotron at the Flerov Laboratory of Nuclear Reactions (FLNR), Joint Institute for Nuclear Research (JINR). The possibility of nanoperforation of GO films in the middle to low electronic stopping power range was examined with 167 MeV Xe, 107 MeV Kr, 61 MeV V, and 46 MeV Ar ions accelerated at the IC-100 cyclotron complex for applied research and industrial applications (FLNR, JINR). The $Se$ values, calculated with SRIM-2013 code, are listed in Supplementary Table 1.

### Characterization

SEM imaging was carried out in secondary electron (SE) mode using a Hitachi SU8020 field emission scanning electron microscope operating at 10 keV. Images of the surface were taken at the surface normal parallel to the electron beam axis (top view) and the surface normal tilted ~45° from the beam axis (tilt view). To reduce charging during SEM imaging, the surface of some specimens was sputter coated with ~5 nm of Pt/Pd (80/20). TEM analysis of 1- to 5-layer-thick GO films deposited on Quantifoil 2/4 200 mesh grids was carried out on a Talos F200 (Thermo Scientific) electron microscope at an operating voltage of 70 keV. The number of layers in the analyzed region was determined based on selected area electron diffraction patterns. EELS spectra were acquired using a Gatan Enfina spectrometer with an energy dispersion of 0.5 eV/channel. Raman spectra were recorded under low power (~100 µW) using a Nanofinder 30 (SOL Instruments) spectrometer with 473 nm laser excitation and a 100× objective lens. The Raman spectra

were background corrected and deconvoluted using a single Lorentzian and Breit-Wigner-Fano line for the D- and G-bands, respectively[42]. FT-IR measurements were carried out on a Nicolet 6700 (Thermo Scientific) spectrometer in the wavelength range of 5000–600 cm$^{-1}$ using a single reflection Smart iTR attenuated total reflection (ATR) accessory with a Ge internal reflection element. The FT-IR spectra were corrected to eliminate the distortion in the relative intensities of bands and shifts in their frequencies by using an advanced ATR correction algorithm as implemented in OMNIC 9.2 software. XPS measurements were performed on a K-Alpha spectrometer (Thermo Scientific). The current–voltage ($I$–$V$) characteristics of GO films were recorded in a two-probe configuration within a voltage range of −1/+1 V using a Lakeshore TTPX probe station equipped with tungsten tips of 25 µm radius and a Keithley 2636B source measure unit.

### MC-RxMD modeling of SHI irradiation damage and structural recovery

The effect of heavy ion passage on the GO structure was modeled with a multiscale hybrid approach consisting of a MC model (TREKIS code[60,61]) for the kinetics of the electronic subsystem and initial lattice excitation and RxMD for subsequent structural transformations[62]. All details on the MC-RxMD modeling are given in Supplementary Note 5.

## Data availability

The data that support the findings of this study are included in the paper and its Supplementary Information and are available from the corresponding author upon request. Data related to the Monte Carlo and molecular dynamics simulations are available from R.A.R. (r.a.rymzhanov@gmail.com). Source data are provided with this paper.

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

## Acknowledgements

The authors are thankful to Dr. O.L. Orelovich (FLNR, JINR), and J. Wółkiewicz (NCU) for recording SEM images and to Dr. A.S. Sohatsky (FLNR, JINR) for TEM analysis. Dr. G. Trykowski and K. Olejniczak (NCU) are thanked for their assistance with EELS measurements. Dr. Ž. Mravik (University of Belgrade) is sincerely thanked for providing the GO-Z samples. Prof. V.A. Skuratov (FLNR, JINR) is thanked for his assistance with heavy ion irradiation. R.A.R acknowledges financial support from the Ministry of Education and Science of the Republic of Kazakhstan (grant number AP09259476). This work used computing resources of GSI (Darmstadt, Germany), the Federal collective usage center Complex for Simulation and Data Processing for Mega-science Facilities at NRC "Kurchatov Institute" (http://ckp.nrcki.ru/), and the computing infra-structure built in the project "PLATON—Science Services Platform" No. POIG.02.03.00-00-028/08.

## Author contributions

A.O. designed and performed experiments, analyzed data and wrote the manuscript. R.A.R. created GO models, performed the MC-RxMD calculations and drafted Supplementary Note 5. All authors discussed the results and commented on the manuscript.

## Competing interests

The authors declare no competing interests.
