## [Peer Review File · Nature Communications]

From nanohole to ultralong straight nanochannel fabrication in graphene oxide with swift heavy ionsREVIEWER COMMENTS

Reviewer #1 (Remarks to the Author):

Andrzej and Ruslan fabricated nanopores (or nanochannels) in graphene oxide (GO) membranes with a series of heavy ions at different energies from 61 MeV to 710 MeV. SEM results show the diameter of the nanopores/nanochannels is $\sim 6-7$ nm with standard deviations of ~ 1 nm. They also combined the FTIR spectrum measurements and MC-RxMD simulations to explore the nanopore formation mechanism. They found that the nanopore formation is dependent on the Se values of the irradiated ions and the chemical composition of GO. This manuscript introduces a new nanopore fabrication method and is well written. However, I would not recommend its publication unless they could clarify the following concerns.

1) One major concern is that the size of the nanopores/channels is too greater for most of the demanding separation applications. Ideally, the channel size for efficient desalting or ionic separations should be around or less than 1 nm. The authors may want to demonstrate the possible applications of their nanochannels.

2) One of the most important questions in nanochannel fabrication is how to tune the channel size. It would be better if the authors could add more discussion addressing this question.

a) It would be better if the authors can investigate the irradiation conditions such as Se values on the nanochannel size, and ideally how to reduce the nanochannels size.

b) The MD simulations suggest that nanochannel size is small at low functional group density. It would be better if the authors could try to confirm this result with irradiation on reduced GO membranes.

c) It is not very clear why the nanopore size in the single-layer GO membrane is larger than that in the multilayered GO membranes. As the authors claimed, the activation agents for etching can easily escape to the vacuum for the single-layer GO membrane. If this is true, the etching of the multilayered GO membranes should be greater and the pore size should be larger.

d) Why the nanopore sizes at the "base" and "at the background level" are different? And why both of them are different from the width of the cross-section of the channel? And how did the authors determine the channel width given such a low contrast of the image?

e) Please explain the pore size difference for the suspended and supported GO films.

f) Explain the pore size difference for GO-Z is resulting from a decreased flake size or the functional group difference.

3) The claim that the new method is scalable in fabrication could be misleading. Although the accelerators have the potential to process a large area of samples, it has yet to demonstrate that scalable GO membranes can be reproduced with a large size comparable to the polymer samples.

4) Show the measured pore density and compare it with the ion fluence to double-check the yield of the nanopore.

5) The authors need to add more information on their simulation model. It would be better if the authors could show the dynamic formation of the nanopores in their MD simulations. And please show evidence that the pore size stabilizes in the simulation. Please explain why "partial defunctionalization occurred for GO cells". Why is there a large gap between the GO sheets? If the direction perpendicular to the GO membrane is not under periodic boundary conditions, how do the authors stabilize the position of GO membranes in this direction?

6) The presentation of the manuscript should be improved to be more readable for the general readership.

P3, Fig.1a, explain what are the yellow dashed lines?

P4, Explain what is "GO-Z" at the first appearance. Explain or illustrate it in the figure "what is their bases and at the background level".

P4, Please show a gray bar of the height for Fig. 2d, it appears the size of all the hillocks is less than 20 nm, why the average diameter is 27 nm?

P4, Fig.2d', what does the yellow part represent?

P4, Show the Se values, i.e., LET (keV/nm), of all the irradiated ions used in this study.

P4 and P5, Please show the raw images for the nanopores at "base" and "at the background level".

P5, Add captions for Figures 3 c' and d'.

P6, It is very hard to follow the FT-IR results. It is better to put the FT-IR spectra of pristine GO-Z and GO-M together to show the band difference; Explain what the yellow boxes are in Fig.4b in the captions; It is very hard to see the data of Bi in Fig. 4c-f; Please mark the positions of D-band and G-band in the spectra.

P8, Fig.4b caption "Raman spectral", where is the data?

P10, "In the first set" actually refers to the second set in Table 1.

P15, "the pores have a perfectly round geometry" is inconsistent with the SEM images shown before.

Reviewer #2 (Remarks to the Author):

This work uses experimental synthesis, characterization, and molecular dynamics simulations to shed light on nanopore formation in graphene oxide. The work is rigorous, the data analysis is sound, and the conclusions are supported by the results presented. The authors demonstrate fabrication of almost uniform 7 nm straight nanochannels in graphene oxide using swift heavy ion irradiation. This work examined two types of flakes, synthesized differently, with different flake sizes, functional groups, and defect concentration. XPS was used to determine the C/O ratio and FT-IR was used to get an understanding of the functionalization. SEM and TEM characterization were used to find out about the nanochannel geometry. The electronic stopping power was varied by changing the ions type and ion energy, for instance 700 MeV Bi vs 167 MeV Xe ions. Reactive molecular dynamics was used to understand the effect of the nature of functional groups and functionalization density on pore formation. The overall results are presented well and there is supporting information in the supplementary material. The manuscript is well written and accessible to a general scientific audience. Prior work is adequately cited. There are no glaring weaknesses discernible.

Given the limitations of the simulation method, the model does not exactly correspond to the experimental results but sheds enough light on the findings. It is not clear if the method can be scaled up economically to fabricate this nanoporous material for applications in selective separations, sensing or energy storage. But that is not a major deficiency of this work.

This is good quality work that is recommended for publication.

Reviewer #3 (Remarks to the Author):

The manuscript, " From nanohole to ultralong straight nanochannel fabrication in graphene oxide with swift heavy ions" reports the synthesis of a porous graphene oxide thin films (also designated as graphene oxide paper, Nature 2007, 448, 457–460) by controlled irradiation with swift heavy ions. This approach allows high control over the pore size of the graphene oxide films in the nanometric range, 6-7 nm-sized holes, pore density, and uniformity. Additionally, it was demonstrated the formation of vertically aligned nanochannels on graphene oxide films with a

thickness of up to 10- μ m, by irradiation with high-energy heavy ions.

The reported work in the present article is interesting and of high relevance to the scientific community. However, the novelty is strongly limited once its proof of concept has been clearly demonstrated by the authors in two previously published articles. By using swift heavy-ion irradiation, the authors already demonstrated that nano-sized pores could be formed on graphene oxide (Carbon, 2019,141, 390–399) and graphene (Nanoscale, 2018,10, 14499–14509).

The abstract is presented in a very clear and objective manner. The introduction is well structured and allows to properly frame the relevance of the obtained results in the current state of the art. The main conclusions of the article are well supported by the presented data, which includes a detailed chemical and structural analysis and a respective correlation with the MC-RxMD modelling of pore formation. However, a clear limitation can be identified in this section that is related to the lack of data that supports the claim that pore periphery is electrically conductive due to the partial restoration of graphitic structure. The manuscript presents significant and updated references.

The article is well structured, and the presented data is concise and well supported by the implemented discussion. Electronic microscopy characterization gives strong proof that the authors were able to control the formation of 6-7 nm-sized holes and ultralong straight nanochannels (Fig 1, 2 and 3). On the basis of FTIR and Raman analysis, the authors also discussed in detail the reduction effects caused by SHI-irradiation on the GO films.. However, it's important to keep in mind that both of these characterization techniques are used for bulk analysis of GO films. Taking this into account, is difficult to infer that the reduction effects observed can be mainly attributed to the frontier regions of the formed holes, as the authors suggested. Importantly, these results should be complemented by a detailed XPS analysis of C1s peak for the different GO films. Additional experiments should be conducted in order to improve the overall quality of the manuscript. Further experimental details should be provided in terms of the two different types of GO (GO-Z and GO-M) used for the preparation of the films. The characterization of graphene-based materials is already well established and is based on three main factors: the number of carbon layers, the C/O ratio, and lateral dimensions (Angew. Chem. Int. Ed. 2014, 53, 2–7). The authors should provide an extensive analysis of the starting material, GO-Z and GO-M, having in consideration these three parameters, by performing detailed XPS analysis (C/O ratio), AFM (thickness of nanosheet), and electronic microscopy (statistical analysis of lateral size distribution).

HRTEM and EELS analysis are considered essential to fully understanding the structure and chemical composition at the frontier regions of the formed holes in GO films (Carbon 2021, 178, 477-487). (Carbon 2021, 178, 477-487). These characterization techniques will enable a local detailed analysis of the structural arrangements on carbon atoms caused by the irradiation of swift heavy ions, as well as access to the dominant chemical bonds, including transformations of different oxygen functional groups and graphitisation as a function of ion fluences.

The MC-RxMD modeling that was used to try to predict pore formation and structural recovery is outside of my area of expertise.

Reviewers response:

The reviewers are thanked for their valuable comments on our manuscript. A point-by-point response is below:

Reviewer #1

1) One major concern is that the size of the nanopores/channels is too greater for most of the demanding separation applications. Ideally, the channel size for efficient desalting or ionic separations should be around or less than 1 nm. The authors may want to demonstrate the possible applications of their nanochannels.

For our samples, we indeed found that pores in this size range (1 nm, or less) can be created in very thin (preferably one layer thick) suspended GO films irradiated with 167 MeV Xe ions, i.e., in the mid electronic stopping power range. These pores are closely similar to those obtained in our realistic SEE-corrected MC-RxMD model of 167 MeV Xe ion-irradiated single layer GO, as shown in Supplementary Fig. 20 a,bottom.

We doubt, however, that these results are in line with the concept of our manuscript. First, these ~ 1 nm-sized pores have irregular shapes and worse size uniformity. Second, for GO, a few excellent approaches have been developed for utilizing and tuning the slit-shaped pores between GO sheets (see, e.g., Ref. 24 and 25, initial submission); these approaches have been proven to be efficient in applications such as desalting and ionic separations.

In contrast, there is a striking lack of serial, scalable methods of creating the pores in the few nm-size range, i.e., as these in our manuscript. Such nanoporated graphene-based films are required for several important applications:

- Biomolecular and biofluidic applications, e.g., molecular sieving, recognition-based sensing, and DNA sequencing – due to the perfect pore geometry, narrow pore size distribution, and presence of nitrile groups in pore walls enabling immobilization of biorecognition molecules;
- Membrane technologies – as novel GO track membranes for ultrafiltration and separation with an easily adjustable permeability-selectivity ratio;
- Electrical energy storage – as an electrode material for high-rate charge/discharge devices – here, the vertically straight nanochannels are supposed to provide high mass transport and shorten the flux pathways that, for typical nanoporated GO films with random orientation of flakes and randomly distributed basal holes, are still undesirably long;
- High-resolution electron microscopy imaging – as holey TEM grids with ultranarrow holes providing unobstructed regions for objects that are too small to be supported on commercial holey grids (holes ~ 100 nm in size).

We listed these examples in our manuscript. We are ready, however, to address this issue in more detail if needed.

Changes in the MS:

Possible applications of the nanochannels are listed (Supplementary Note 1).

The statement: “Xe ions of 167 MeV energy ... cannot “drill” nanoholes in suspended single- to few-layer GO” was misleading and has been replaced by: “Xe ions of 167 MeV energy ... cannot “drill” nanoholes in unfolded, suspended few-layer GO”.

The statement on SEE-corrected results has been corrected and now reads: “The obtained results ... show that the reduction in energy deposition due to SEE is high enough to reduce the damage in the track core to the extent that only small pores, 0.4-0.9 nm in size, are created.”

2) One of the most important questions in nanochannel fabrication is how to tune the channel size. It would be better if the authors could add more discussion addressing this question.

a) It would be better if the authors can investigate the irradiation conditions such as Se values on the nanochannel size, and ideally how to reduce the nanochannels size.

We expect the nanochannel width to be reduced with decreasing Se value. This behavior is already visible when comparing the channel sizes for Bi (6.6-7 nm) and Xe-ion (4-5.2 nm) irradiated films. However, we also expect a certain Se threshold, below which only small irregularly distributed defects will be created instead of the channels. This means that for the present samples, the expected channel width can be tuned only in a few nm range. Based on the surface SEM images, we suppose that the channels with widths smaller than those for Xe- and Bi-ion-irradiated samples were also created in GO films irradiated with 107 MeV Kr ions (mid Se range) but were absent in samples irradiated with low Se ions (V and Ar). We are ready to address this issue in more detail; however, we would like to stress that imaging such small features is extremely difficult with SEM microscopy.

Changes in the MS: The following phrase was included: Cross-sectional SEM images (Fig. 3c,c') show that the channels are ca. 4-5.2 nm in diameter, i.e., their widths were reduced compared to those formed in 710 MeV Bi ion irradiated films.

b) The MD simulations suggest that nanochannel size is small at low functional group density. It would be better if the authors could try to confirm this result with irradiation on reduced GO membranes.

The reviewer is thanked for this very interesting idea. We performed preliminary experiments for the initial GO films and those thermally treated under mild conditions. Such treatment conditions are known to be sufficient to remove the most unstable oxygen functional groups. The GO films tested here were obtained according to the same modified Hummers method as GO-Z but have different initial C/O ratios. The samples were irradiated with 167 MeV Xe ions to different fluences, and small angle X-ray scattering (SAXS) patterns were recorded. The fitting of SAXS data proved that cylinders (representing the nanochannels) formed in thermally treated samples indeed have a smaller radius than those created in nontreated GO films. We have included some preliminary results as Related Manuscript File for review purposes.

Since the topic of reduced graphene oxide goes well beyond the scope of our manuscript and we have already reached the word count limits, we would like to retreat from including these data in the manuscript.

c) It is not very clear why the nanopore size in the single-layer GO membrane is larger than that in the multilayered GO membranes. As the authors claimed, the activation agents for etching can easily escape to the vacuum for the single-layer GO membrane. If this is true, the etching of the multilayer GO membranes should be greater, and the pore size should be larger.

The reviewer is thanked for this comment; we centrally have not been clear enough with our description. We want to highlight that the scenarios for single-, few-, and multilayer GO are quite different. To address this issue in detail, we performed additional simulations for $C_6H_{1.5}O_{1.8}$ structures with different numbers of layers. We chose this particular composition, since the cracks that appeared in the hole edge in the simulations (Fig. 5a) closely resemble those observed in TEM images of single-layer GO (Fig 1a). In the case of single-layer GO, upon ion impact, the lattice undergoes significant distortion (see Supplementary movie 1), increasing the probability of crack formation on the hole edge. For few-layer GO, the internal layers are stabilized by the outer layers (see Supplementary Movie 2), diminishing the strain and edge tearing. Lastly, for thick GO films, the reactive intermediate species cannot that easily outdiffuse to vacuum and each channel walls, as can be clearly seen in Supplementary Move 3.

Changes in the manuscript:

This different behavior for single-, few- and multilayer structures is now explained, and the results of additional simulations are discussed in detail (page 15). Several minor improvements in the text on page 3 (paragraph: High-electronic-stopping-power ions) were done. New Supplementary Figure 18 was included.

d) Why the nanopore sizes at the “base” and “at the background level” are different? And why both of them are different from the width of the cross-section of the channel? And how did the authors determine the channel width given such a low contrast of the image?

The reviewer is thanked for pointing out the lack of description of our approaches to the determination of pore size parameters from microscopy images. Since the contrast in most of our images is rather low, the determination of pore parameters from top-view images was based on radial (i.e., angularly averaged) intensity profiles and that from cross-sectional images employed aligning and averaging of line profiles – see Supplementary Note 2.

The meaning of nanopore sizes “at the base” and “at the background level” is now explained (Supplementary Note 2 and Supplementary Figure 2). Since the channel is broader at the entrance (as also confirmed by MC-RxMD modeling, Fig. 2d,d’), the values of pore size “at the base” are higher than those “at the background level” (see Supplementary Figure 2). The pore sizes “at the background level” should more closely match the channel width determined from cross-sectional images. The discrepancy is mainly due to the fact that we are approaching the resolution limit of our SEM microscope; in the collected images, these features are only a few pixels in width.

Changes in the manuscript: The methodology is now explained in Supplementary Note 2 and referred to in the manuscript (Fig. 2c’, caption).

e) Please explain the pore size difference for the suspended and supported GO films.

We addressed this issue in our simulations – supported GO film can be treated as multilayer structure – please see our response to your issue raised in point 2c.

f) Explain the pore size difference for GO-Z is resulting from a decreased flake size or the functional group difference.

As we mentioned in the manuscript, there are reasons that both factors (i.e., flake size and group type composition) can influence the pore size. In GO, the functional groups that occupy basal planes are different than those that saturate graphene edges. As a result, the group type distribution and flake size distribution are mutually related; this makes it impossible to experimentally investigate which of these two parameters exerts a dominant role on the pore sizes.

3) The claim that the new method is scalable in fabrication could be misleading. Although the accelerators have the potential to process a large area of samples, it has yet to demonstrate that scalable GO membranes can be reproduced with a large size comparable to the polymer samples.

We understand the reviewer's concerns about large-scale fabrication. In response, we would like to present an outline of a straightforward implementation of scalable fabrication of hybrid membranes with SHI-patterned GO thin films (selective layer) supported on porous polymer substrates (passive layer). Realization of such an approach is possible thanks to the development of a few scalable fabrication techniques of uniform large-area GO coatings on flexible substrates (e.g., polymers). Among these methods, doctor blade coating, slot-die coating, bar coating and its recent modification based on a Mayer rod, yield coatings with aligned GO layers, i.e., paper-like structure similar to that of the samples used in the present study. Importantly, these coating techniques are capable of continuous roll-to-roll processing, which is the main processing technology used in cyclotron-based large-scale production (e.g., of polymer track membranes).

We would like to mention that our samples shown in Fig. 3b-d' are GO films deposited on the Nylon-6 polymeric support. Nylon-6 is known to provide high adhesion to the GO layer and can be obtained with pores large enough to provide undisturbed flow in separation oriented applications. Note that in Fig. 3c' and d', the patterned GO (cross-sectional and bottom views) is suspended over a large pore in Nylon. A low magnification cross-sectional image showing the structure of such a hybrid membrane is now included in the new Supplementary Fig. 21.

Changes in the MS: In Supplementary Fig. 21, we present a general scheme for manufacturing hybrid GO-based track membranes. In the figure caption, we summarize the general steps used in processing. We also clarified the caption of Fig. 3b-d' to emphasize that the patterned GO layer is supported.

4) Show the measured pore density and compare it with the ion fluence to double-check the yield of the nanopore.

We examined all samples irradiated with 710 MeV Bi ions, both suspended thin films and supported thick layers, and found that the nanoholes/channels are created with 100% efficiency.

5) The authors need to add more information on their simulation model. It would be better if the authors could show the dynamic formation of the nanopores in their MD simulations. And please show evidence that the pore size stabilizes in the simulation. Please explain why “partial defunctionalization occurred for GO cells”. Why is there a large gap between the GO sheets? If the direction perpendicular to the GO membrane is not under periodic boundary conditions, how do the authors stabilize the position of GO membranes in this direction?

We split the Reviewer’s comment into 5 subsections denoted (i) to (iv):

(i) The authors need to add more information on their simulation model.

Additional information on MC-RxMD modeling is now included in Supplementary Note 5 (please see the text in blue). In particular, for MC calculations, we explained (i) the Born approximation toward linking the loss function with the cross-section of scattering of an incident particle on a spatially and dynamically coupled system of scattering centers and (ii) reconstruction of the loss function from the experimental and calculated optical data (by the Ritchie and Howie algorithm). In Supp. Table 3 we listed the values of the coefficients of the energy loss function in the form of optical oscillators. We also showed a comparison of the initial energy deposition into the GO lattice for 167 MeV Xe and 710 MeV Bi ions (Supp. Fig. 5b). If the reviewer feels that any other details should be specified, we will be glad to follow the reviewer’s suggestions.

(ii) It would be better if the authors could show the dynamic formation of the nanopores in their MD simulations. And please show evidence that the pore size stabilizes in the simulation.

The dynamics of nanopore formation have now been shown for the following GO structures:

- single layer ($C_6H_{1.2}O_{2.4}$, tilted perspective view),
- 3-layer ($C_6H_{1.5}O_{1.8}$, cross-sectional view),
- multilayer model consisting of 10 layers ($C_6H_{1.2}O_{2.4}$, cross-sectional view).

To better visualize the initial stage of nanopore formation, the first few ps of the moves for $C_6H_{1.2}O_{2.4}$ (perspective view) and the multilayer model (cross-sectional view) are shown in slow motion. After that, to show that the size of the nanopore stabilizes, the animations continue in a normal frame rate to the end time of 100 ps. We consider these animations as evidence for nanopore size stabilization; however, upon the Reviewer request, we are ready to calculate time-dependent changes in the radial distribution profiles or radii of the cylindrical vdW probe fitted to the nanohole.

Changes in the MS: New supplementary movies were included:

Supplementary Movie 1: Dynamics of nanopore formation in single layer GO ($C_6H_{1.5}O_{1.8}$, tilted perspective view). File: go_C6H1.5O1.8_1layer_animation_tach.mp4

Supplementary Movie 2: Dynamics of nanopore formation in 3-layer GO ($C_6H_{1.5}O_{1.8}$, cross-sectional view). File: go_C6H1.5O1.8_3layers_animation_slice_tach.mp4

Supplementary Movie 3: Dynamics of nanopore formation in multilayer GO (10 layers, $C_6H_{1.2}O_{2.4}$, cross-sectional view). File: go_C6H1.2O2.4_10layers_animation_slice_tach.mp4

We also included information (in Supplementary Note 5) that the simulation time of 0.1 ns was sufficient for the pore size to stabilize.

(iii) Please explain why “partial defunctionalization occurred for GO cells”.

We distributed oxygen functional groups in a semirandom manner (see Supplementary Note 5), which resulted, especially for heavy substituted GO lattices, in the appearance of misarranged sites. During the equilibration, these sites relaxed with structural rearrangement (see Supplementary Note 5) accompanied by possible detachment of excessive oxygen-containing species. The validity of such an approach was demonstrated in Liu et al. (Ref. 49, Carbon 2019, 143, 566–577), who showed that the equilibration of the structures with the initial random spatial distribution of functional groups leads to realistic GO models, reflecting the composition of real samples.

(iv) Why is there a large gap between the GO sheets?

Periodic boundary conditions along the xy plane in combination with insufficiently large xy cell dimensions reflect an unrealistic situation in which several ions simultaneously pass the GO flake in closely positioned points (small interimpact distances). For the multilayer model, the evolution of molecular fragments was the most intense; at some point in time, a complete detachment of some GO sheets from the others occurred. Because of the structural distortion (spatial hindrance), the self-alignment occurring in the subsequent stage was only partial. Note that in the irradiation experiment, the ion impacts occur one-by-one and in spatially distant positions.

To get closer to real conditions, we repeated the calculations for a larger simulation cell (> 400k atoms) and found the debonding effect to be neglected.

Changes in the MS: We replaced the old data in Fig 2d and d' with the new simulation results and altered the simulation details accordingly (Supplementary Note 5, lines 223-227). We also included the movie for the new simulation.

(v) If the direction perpendicular to the GO membrane is not under periodic boundary conditions, how do the authors stabilize the position of GO membranes in this direction?

In GO paper, individual sheets are assembled (i.e., tiled) in a near-parallel manner and hold together due to hydrogen bonding interactions and π - π stacking of the defunctionalized islands. The reactive ReaxFF potential accounts for these types of interactions so that the few- and multilayer GO structures are self-stabilized due to interplane interactions. For example, the multilayer model was designed with an initial interlayer distance of 8 Å; during equilibration, the structure undergoes self-assembly with reduction of the spacing to 6.6 Å. No artificial stabilization, such as fixing the positions of the atoms located close to the xy cell boundaries, was used.

Changes in the manuscript: Supplementary Note 5 was supplemented with the following explanation: “This self-assembly was driven by interplane interactions, mainly hydrogen bonding interactions. No artificial stabilization of the modeled structure in the simulation box was applied.”

P3, Fig.1a, explain what are the yellow dashed lines?

We explained this. The dashed circles mark the positions of the nanopores.

P4, Explain what is “GO-Z” at the first appearance. Explain or illustrate it in the figure “what is their bases and at the background level”.

We fixed it. Thank you!

P4, Please show a gray bar of the height for Fig. 2d, it appears the size of all the hillocks is less than 20 nm, why the average diameter is 27 nm?

The image in Fig. 2d' is shown on the same scale as for Fig. 2d. We explained this. Thank you. The diameter of 27 nm results from the 3σ criterion (see Supplementary Note 2). These values can be easily recalculated to a 2σ width (here, 18 nm), which better matches what can be obtained with the naked eye.

P4, Fig.2d', what does the yellow part represent?

We explained this: The bent surface layer of the GO film is marked with orange dashed lines.

P4, Show the Se values, i.e., LET (keV/nm), of all the irradiated ions used in this study.

We apologize for this omission; the Se values, calculated with SRIM-2013 code, are now included in Supplementary Table 1 and referenced on P4.

P4 and P5, Please show the raw images for the nanopores at “base” and “at the background level”.

The picture showing how the pore sizes at “base” and “at the background level” are calculated is now included (Supplementary Fig. 2). Our approach is explained in the response to Reviewer’s comment 2d and in the Supplementary Note 2.

P5, Add captions for Figures 3 c' and d'.

We fixed this. Thank you!

P6, It is very hard to follow the FT-IR results. It is better to put the FT-IR spectra of pristine GO-Z and GO-M together to show the band difference; Explain what the yellow boxes are in Fig.4b in the captions; It is very hard to see the data of Bi in Fig. 4c-f; Please mark the positions of D-band and G-band in the spectra.

In Fig. 4 b, we intended to show the changes in FT-IR spectra, separately for GO-Z and GO-M, occurring upon irradiation over a broad range of ion fluences. For a better comparison of the FT-IR data of the pristine GO-Z and GO-M samples, we included an additional plot (Supplementary Fig. 7) with overlaid spectra in the low wavenumber region ($1700-1150\text{ cm}^{-1}$).

The orange boxes mark the spectral region of conjugated $\text{C}\equiv\text{C}$ bond stretching. We improved the figure caption accordingly.

Because of the limited access to the 710 MeV Bi ion beam, we irradiated the samples with low fluences only. We regret to inform that the Raman data for high-dose irradiation are unavailable.

P8, Fig.4b caption “Raman spectral”, where is the data?

In Fig. 4, we show the parameters obtained from the deconvolution of Raman spectra using Lorentzian and Breit-Wigner-Fano lines. We collected at least 6 spectra for 2 types of specimens irradiated with different fluences of 4 different ions. Such amount of data is hard to present, but can be made available upon Reviewer request.

P10, “In the first set” actually refers to the second set in Table 1.

We fixed this. Thank you!

P15, “the pores have a perfectly round geometry” is inconsistent with the SEM images shown before.

This statement is true for suspended 3-4-layer-thick GO, as shown by TEM. We agree with the Reviewer that in the case of thick GO films (SEM image, Fig. 2c), the pore/channel entries are distorted, but the channels themselves are round.

Reviewer #2

It is not clear if the method can be scaled up economically to fabricate this nanoporous material for applications in selective separations, sensing or energy storage.

The reviewer is thanked for the comments on our manuscript and the recommendation. In the response to the raised issue, in Supplementary Fig. 21 we presented a scalable roll-to-roll fabrication of hybrid GO-based track membranes. In our opinion the proposed approach provides a straightforward and economically effective implementation of our ideas.

Reviewer #3

1. The reported work in the present article is interesting and of high relevance to the scientific community. However, the novelty is strongly limited once its proof of concept has been clearly demonstrated by the authors in two previously published articles. By using swift heavy-ion irradiation, the authors already demonstrated that nano-sized pores could be formed on graphene oxide (Carbon, 2019,141, 390–399) and graphene (Nanoscale, 2018,10, 14499–14509).

In the first mentioned paper (Carbon, 2019,141, 390–399), based on indirect experimental evidence, such as the formation of sp-hybridized chains (via Raman spectroscopy) and changes in friction in the track core regions (via lateral force microscopy), we postulated the possibility of nanohole creation in GO sheets upon irradiation with ions with sufficiently high Se values. The paper focuses on localized reduction of GO with SHI irradiation; the hypothesis of nanohole creation has not been proven in direct experiments. In the second paper (Nanoscale, 2018,10, 14499–14509), we experimentally showed that SHI irradiation of exfoliated multilayer graphene on SiO₂/Si substrate results in the formation of relatively large (20–60 nm) nanoholes, but these nanoholes were created only in the upper one or two layers of the multilayer graphene films; their formation efficiency was low (1 pore per 10 ion impacts), and the creation mechanism still remains puzzling. These nanoholes, owing to their partially reconstructed edges, allowed us to tune the electrical transport properties of the graphene films in an interesting way, but due to the above characteristics (nonthrough, large in diameter), they are impractical and not intended to be used in energy storage and separation-oriented technologies.

We want to stress that the fabrication of GO films with vertically oriented long nanochannels (i.e., GO-based track membranes) is an original concept developed within the present manuscript and has neither been introduced nor demonstrated in our previous papers and known papers by other authors.

2a. The main conclusions of the article are well supported by the presented data, which includes a detailed chemical and structural analysis and a respective correlation with the MC-RxMD modelling of pore formation. However, a clear limitation can be identified in this section that is related to the lack of data that supports the claim that pore periphery is electrically conductive due to the partial restoration of graphitic structure.

We are thankful for this comment - since the unique features of swift heavy ions, such as the ability to modify the materials in a localized way, may not be known to a broad readership of the journal, we feel that the issue requires a brief explanation in the MS.

The basic explanation assumes that the energy initially deposited by the ion to the electron subsystem is then transferred (via electron-phonon coupling) to the lattice atoms, giving rise to a strong local heating of the GO lattice. The temporary temperature increase is the highest in the track core (i.e., the cylindrical region closest to the ion trajectory) and is sufficient to cause damage or etching of the nanoholes/nanochannels. In the track periphery (so-called track halo), lattice heating is not as strong and leads to partial defunctionalization of GO. The last process is similar to well-studied thermal defunctionalization (reduction), in which a partial restoration of graphitic structure occurs, except that it takes place locally, in the frontier region of the nanohole (high Se ions) or inside nanosized spots (low Se ions).

Changes in the Manuscript: The following phrase was added on p. 11: “Such local structural restoration occurs thanks to the unique ability of swift heavy ions to modify the materials within the nanometer-sized cylinders around the ion trajectory (Supplementary Fig. 12)”. The caption of Supplementary Fig. 12. (core-shell model) was improved.

2b. On the basis of FTIR and Raman analysis, the authors also discussed in detail the reduction effects caused by SHI-irradiation on the GO films.. However, it's important to keep in mind that both of these

characterization techniques are used for bulk analysis of GO films. Taking this into account, it is difficult to infer that the reduction effects observed can be mainly attributed to the frontier regions of the formed holes, as the authors suggested.

We completely agree with the Reviewer that the techniques used (FTIR, Raman) provide information on the global (averaged) composition of the films, this is because the lateral resolution is not sufficient to directly prove the existence of nanomodified regions. We want to stress, however, that swift heavy ions have a unique ability to modify the materials in a localized way, i.e., inside the nanosized cylindrical zones around the ion trajectory (please see our response to your comment 2a). The localized character of SHI-induced material modification can be indirectly confirmed by examining the dose dependency of the spectral parameters. Based on optical spectroscopy and X-ray diffraction data, such an approach was successfully used to study localized transformations and damage in a variety of materials. Regarding graphene oxide, it was previously shown that XPS-derived C/O ratios follow the direct impact model, indicating localized defunctionalization inside the tracks (Ref. 44). In Ref. 45, it was shown that Raman data can be fit to a double-exponential model with annealing (i.e., ordering) and damaging cross-sections, which is a signature of structural damage in the track core and recovery in the track halo. Finally, such behavior is also expected from strong heating of the GO lattice in the ion track, confirmed from our MC-RxMD modeling.

Changes in the manuscript: We highlighted the ability of swift heavy ions to create nanomodified regions in the materials.

3. Importantly, these results should be complemented by a detailed XPS analysis of C1s peak for the different GO films.

In the revision, we included C1s core level spectra of both initial films and those irradiated with various fluences of 167 MeV Xe ions. In the analysis, we used the recently developed Doniach-Sunjic-Shirley lineshape model (Surf Interface Anal. 2022;54:67–77) that adequately describe the asymmetry of the sp^2 C peak and overcome numerical obstacles related to the well-established Doniach-Sunjic line.

These results are important in the sense that they (i) confirm the compositional differences of the initial films deduced from FT-IR data, in particular the difference in the content of edge and basal plane groups, (ii) show that the films progressively defunctionalize (i.e., reduce) upon ion bombardment, and (iii) prove that the films irradiated even to high fluences (1×10^{13} ions/cm²) still contain residual amounts of oxygen, in agreement with MC-RxMD simulations.

We want to stress that in terms of probing area, XPS measurements should be categorized in the same way as FTIR and Raman, i.e., as giving average surface composition.

Changes in the manuscript: New Supplementary Figure 8 was included. Several minor changes in the text on p. 7 were made.

4. Further experimental details should be provided in terms of the two different types of GO (GO-Z and GO-M) used for the preparation of the films. The characterization of graphene-based materials is already well established and is based on three main factors: the number of carbon layers, the C/O ratio, and lateral

dimensions (Angew. Chem. Int. Ed. 2014, 53, 2–7). The authors should provide an extensive analysis of the starting material, GO-Z and GO-M, having in consideration these three parameters, by performing detailed XPS analysis (C/O ratio), AFM (thickness of nanosheet), and electronic microscopy (statistical analysis of lateral size distribution).

The reviewer is thanked for this suggestion. For the present study, we chose GO-M and GO-Z samples with similar C/O ratios. Now, we specified the exact value of the C/O ratio, as determined from XPS measurements.

We thoughtfully analyzed the lateral sizes of the GO flakes for both GO-Z and GO-M based on SEM imaging. These measurements were supplemented with statistical analysis and confirmed the assumptions on flake size differences as stated in the initial submission.

Taking into account the difficulty of the AFM technique with scanning an area large enough to obtain statistically meaningful results (see, e.g., Sci. Rep. 2021, 11, 15969) and the fact that AFM images of GO-Z were already reported by Dr. Z. Mravik (Tehnika Novi Mater. 2018, 27, 186-191), we decided to skip the AFM analysis. The AFM measurements by Dr. Z. Mravik (Tehnika Novi Mater. 2018, 27, 186-191), who kindly donated us the GO-Z sample, show that GO-Z consists mainly of monolayer GO sheets. This observation is confirmed by our SEM analysis, based on a light gray color (weak contrast) found for most of the flakes. Similarly, for the GO-M specimen, a conclusion on the content of monolayer flakes was drawn from the analysis of the contrast in the SEM images.

Changes in the manuscript: Supplementary note 4 on flake size determination was significantly improved. The following statement was included (p. 17): “The obtained samples were characterized with respect to the C/O ratio, number of carbon layers and lateral dimensions, according to the recommendations in [Angew. Chem. Int. Ed. 2014, 53, 2–7]”

5. HRTEM and EELS analysis are considered essential to fully understanding the structure and chemical composition at the frontier regions of the formed holes in GO films (Carbon 2021, 178, 477-487). (Carbon 2021, 178, 477-487). These characterization techniques will enable a local detailed analysis of the structural arrangements on carbon atoms caused by the irradiation of swift heavy ions, as well as access to the dominant chemical bonds, including transformations of different oxygen functional groups and graphitisation as a function of ion fluences.

The reviewer is thankful for suggestions to resolve these deficiencies. We acquired high magnification TEM images of suspended thin GO film irradiated with 710 MeV Bi ions (Supplementary figure 13a). The image reveals a structural modification of GO at the frontier regions of the formed holes. The changes are especially visible in regions with densely spaced nanoholes, where consecutive ion impacts in nanohole proximity contribute to stronger lattice modification (i.e., defunctionalization). Based on our MC-RxMD modeling, lattice heating in the nanohole periphery is expected to lead to a partially recovered graphitic structure with some defects. Because of the hybridization change from sp^3 to sp^2 , the defunctionalized regions are flatter, enabling better imaging of the graphitic planes.

Regarding EELS investigations, Pelaez-Fernandez et al. (Carbon 2021, 178, 477-487) in their detailed work averaged spectra from large-area scans (~ 150 nm in diameter). Such an approach was hardly possible in our case, since the sizes of the fabricated features are much smaller. As is known, the structure of carbon nanomaterials, such as GO, can be easily affected by excessive exposure to electron beams during TEM/EELS experiments. Bearing that in mind, we collected the carbon K-edge spectra with low acquisition time and lower dispersion per channel (i.e., lower spreading of the spectrum signal over a CCD detector matrix). Under these conditions, spectra measured on an unmodified region of the sample are similar to those reported previously for GO (e.g., [D. D'Angelo Carbon 93 (2015) 1034–1041, and S. Hettler, 2D Mater. 8 (2021) 031001]); however, the signals of particular functional groups are not as well resolved as in Pelaez-Fernandez et al. The spectrum measured on a defunctionalized region reveals signatures of partial recovery of the graphitic structure around the nanoholes (see Supplementary figure 13b). We want to stress that such measurements on nanoscale regions are particularly difficult; because of the sample drift, the spectrum recorded around the nanohole is possibly “contaminated” by a signal from unmodified GO.

Changes in the manuscript: New Supplementary Figure 13 was included. The following phrase was added to the main body of the MS: “The partial recovery of the graphitic structure at the frontier regions of the created nanoholes is confirmed by TEM and electron energy loss spectroscopy (EELS), Supplementary Fig. 13.”

REVIEWERS' COMMENTS

Reviewer #1 (Remarks to the Author):

The authors have addressed all my concerns in this revision, I would recommend its publication.

Reviewer #2 (Remarks to the Author):

This manuscript has been revised in response to the comments provided during the previous review. This work has used swift heavy ion irradiation to create nano holes and nano channels in graphene oxide without subsequent chemical treatment or post irradiation processing. In addition to synthesis and irradiation, a variety of characterization techniques and simulations have been used. The results show the promise of developing this material system for selective separations and sensing. Questions still remain about how realistic it is to scale up this method to commercial/industrial processes. The work is original and builds upon the literature including prior work by this group. The methodology, conclusions and data analysis are sound. The supplementary material is adequate. The authors have addressed the issues raised by this reviewer. The revised manuscript is recommended for publication. Despite lingering questions about the feasibility of scaleup and the economics of this process, the work is a valuable addition to the literature.

Reviewer #3 (Remarks to the Author):

The authors provided the appropriate answers to most of my previous comments related to the scientific work presented in the first version of the article. However, the novelty of the work for a communication is somewhat limited by the authors previous articles published.

Reviewer #1 (Remarks to the Author):

The authors have addressed all my concerns in this revision, I would recommend its publication.

We appreciate the reviewer's recommendation. We would also like to thank the reviewer for careful reading of the manuscript and valuable comments allowing us to substantially improve our study.

Reviewer #2 (Remarks to the Author):

This manuscript has been revised in response to the comments provided during the previous review. This work has used swift heavy ion irradiation to create nano holes and nano channels in graphene oxide without subsequent chemical treatment or post irradiation processing. In addition to synthesis and irradiation, a variety of characterization techniques and simulations have been used. The results show the promise of developing this material system for selective separations and sensing. Questions still remain about how realistic it is to scale up this method to commercial/industrial processes. The work is original and builds upon the literature including prior work by this group. The methodology, conclusions and data analysis are sound. The supplementary material is adequate. The authors have addressed the issues raised by this reviewer. The revised manuscript is recommended for publication. Despite lingering questions about the feasibility of scaleup and the economics of this process, the work is a valuable addition to the literature.

We sincerely thank the reviewer for valuable comments and the recommendation.

Expanding to commercial/industrial fabrication typically requires enormous amount of work. As an example, we would like to mention polymer track-etched membranes with magnetron sputtered Ti/TiO₂ layers, developed by our group. Here, scaling up from laboratory process (Thin Solid Films 2021, 725, 138641) to roll-to-roll fabrication, involved a year-long study to overcome scientific- and engineering-related obstacles (for details see: Surfaces and Interfaces 2022, 31, 101975).

Such specific technically oriented investigations, first, go beyond the scope of our manuscript, and second, might be of limited interest of the readership of the

NCOMMS journal. We believe, however that such scaling up, especially for polymer-supported GO films is possible (as outlined in Supplementary Fig. 22). Based on the application potential of such materials, we plan to address this issue in our further research.

Reviewer #3 (Remarks to the Author):

The authors provided the appropriate answers to most of my previous comments related to the scientific work presented in the first version of the article. However, the novelty of the work for a communication is somewhat limited by the authors previous articles published.

Once again we thank the reviewer for constructive comments and suggestions of additional experiments that allowed to improve our study.

In our previous papers, we focused on different aspects of ion beam modification of graphene and graphene oxide, mainly electrical transport properties. We want to once again stress, that fabrication of long vertically-oriented nanochannels in GO films as well as nanoholes in suspended GO layers are new developments that have not been demonstrated in our previous articles.